# Temperature dependence of the Brewer global UV measurements

Ilias Fountoulakis [1], Alberto Redondas[2], Kaisa Lakkala[3,5], Alberto Berjon [4], Alkiviadis F. Bais[1], Lionel Doppler[5], Uwe Feister[5], Anu Heikkila[6], Tomi Karppinen[3], Juha M. Karhu[3], Tapani Koskela[6,*], Katerina Garane[1], Konstantinos Fragkos[1,7], and Volodya Savastiouk[8]

[1]Aristotle University of Thessaloniki, Laboratory of Atmospheric Physics, Thessaloniki, Greece
[2]Agencia Estatal de Meteorología, Izaña Atmospheric Research Center, Spain
[3]Finnish Meteorological Institute – Arctic Research Centre, Sodankylä, Finland
[4]University of La Laguna, Department of Industrial Engineering, S.C. de Tenerife, Spain
[5]Deutscher Wetterdienst, Meteorologisches Observatorium Lindenberg – Richard Assmann Observatorium (DWD, MOL-RAO)
[6]Finnish Meteorological Institute, Climate Research, Helsinki, Finland
[7]National Institute of R&D for Optoelectronics, Magurele, Romania
[8]International Ozone Services Inc, Toronto, Canada
[*]works now as independent researcher

*Correspondence to:* Alberto Redondas (aredondasm@aemet.es)

**Abstract.** Spectral measurements of global UV irradiance recorded by Brewer spectrophotometers can be significantly affected by instrument-specific optical and mechanical features, thus proper corrections are needed in order to reduce the associated uncertainties to within acceptable levels. The present study aims to contribute to the reduction of uncertainties originating from changes in the Brewer's internal temperature, which affect the performance of the optical and electronic parts, and subsequently the response of the instrument. Until now, measurements of the irradiance from various types of lamps at different temperatures are used to characterize the instruments' temperature dependence. The use of 50 Watt lamps was found to induce errors in the characterization due to changes in the transmissivity of the Teflon diffuser as it warms up by the heat of the lamp. In contrast the use of 200 or 1000 Watt lamps is considered more appropriate because they are positioned at longer distances from the diffuser so that warming is negligible. Temperature gradients inside the instrument can cause mechanical stresses which can affect the instrument's optical characteristics. Therefore during the temperature-dependence characterization procedure warming or cooling must be slow enough to minimize these effects. In this study, results of the temperature characterization of eight different Brewer spectrophotometers operating in Greece, Finland, Germany and Spain are presented. It was found that the instruments' response is changing differently in different temperature regions due to different responses of the diffusers' transmittance. The temperature correction factors derived for the Brewer spectrophotometers operating at Thessaloniki, Greece and Sodankylä, Finland were evaluated and were found to remove the temperature dependence of the instruments' sensitivity.

## 1 Introduction

The Brewer spectrophotometer was developed in the 1970s and became commercially available in the beginning of the 1980s (Brewer, 1973; Kerr et al., 1985b). They were initially designed to measure the total columns of ozone ($O_3$) and sulfur dioxide

($SO_2$) and, towards the end of the 1980s, they were modified to perform also spectral measurements of the global solar UV irradiance (Kerr and McElroy, 1995; Bais et al., 1996). Nowadays, more than 200 Brewer spectrophotometers are deployed worldwide. Spectral UV measurements from Brewers have been used in a number of important studies which highlighted the impact of the stratospheric ozone depletion until the mid-1990s on the levels of the solar UV-B irradiance that reaches the
earth surface (Fioletov et al., 2001; Kerr and McElroy, 1993; Lakkala et al., 2003; McKenzie et al., 1999) and quantified the interaction between the solar UV irradiance, the earth surface and the atmospheric components which mainly control its levels, such as ozone, sulfur dioxide, aerosols and clouds (e.g. Arola et al., 2003; Bais et al., 1993; Bernhard et al., 2007; Fioletov et al., 1998). Spectral measurements from Brewers have been used widely for climatological studies of biologically effective UV doses (e.g. Fioletov et al., 2003, 2009; Kimlin, 2004), validation of satellite products (e.g. Arola et al., 2002; Bernhard
et al., 2015; Kazadzis et al., 2009) and validation of radiative transfer models (Kazantzidis et al., 2001; Mayer et al., 1997). Lately, spectra from stations with long measurement records have been used for the study of the changes of the solar UV irradiance, showing that changes in air quality and climate have an important impact on its short- and long-term variability (De Bock et al., 2014; Fountoulakis et al., 2016a; Fragkos et al., 2015; Lakkala et al., 2017; Simic et al., 2011; Smedley et al., 2012; Zerefos et al., 2012).

The uncertainty in the measurements of the total ozone column (TOC) is considered to be low, of the order of 1% (Kerr et al., 1985a), while the uncertainty in the measurements of the global spectral UV irradiance for wavelengths greater than 305 nm is estimated to be less than ∼6.5% for well maintained and calibrated instruments (Bernhard and Seckmeyer, 1999; Garane et al., 2006). However, insufficient correction for the effects of individual constructional and operational characteristics, e.g. stray light (Karppinen et al., 2014), dead time (Fountoulakis et al., 2016b), cosine response (Antón et al., 2008; Bais et al.,
1998) and temperature dependence (Garane et al., 2006; Lakkala et al., 2008), may lead to even larger uncertainties (Gröbner et al., 2006). Thus, better understanding of the instrument's characteristics and improvement of the characterization methods are necessary for keeping the uncertainties within acceptable limits (Seckmeyer et al., 2001). Improvement of the quality of the spectra is also essential for the detection of trends in the time-series of the measured irradiance (Weatherhead et al., 1998).

Changes in the internal temperature can affect the electronic, mechanical and optical parts, and subsequently the spectral
response of each individual Brewer spectrophotometer (Kerr, 2010). They have multiple and complex effects on the spectral response of the Brewer spectrophotometers. Existing studies (e.g. Garane et al., 2006; Lakkala et al., 2008; Weatherhead et al., 2001) suggest that temperature mainly affects the response of the photomultiplier tube (PMT) and the transmittance of the $NiSO_4$ filter used in the single monochromator Brewers. However, it is also possible that temperature affects the transmittance of the Teflon diffuser located at the entrance of the fore-optics (Ylianttila and Schreder, 2005) and also causes subtraction and
contraction of the instruments' mechanical components which may affect their response. The characterization procedure (and the subsequent correction of the spectra for the effects of temperature) is quite difficult and uncertain for the following reasons: (1) Different components of the instrument are affected differently by changes in temperature, (2) specific components (e.g. the PMT, the standard lamp or the heater) increase the temperature locally while they are operating, resulting in large temperature gradients inside the instrument, (3) the characterization conditions (e.g. warming and cooling rate) differ from the conditions during the instrument regular operation, (4) the lamps used for the characterization may warm the diffuser and (5) the effects

of temperature depend on the individual characteristics of each instrument. Nevertheless, proper characterization is necessary in order to take the effects of temperature into account and avoid errors in the final products.

The present study is focused on the evaluation of the characterization and correction methods for the effects of temperature on the measurements of the global UV irradiance. Evaluation of the currently used methodology for the characterization and correction of the direct irradiance and TOC measurements is out of the scope of the present study. The internal temperature of the Brewer may change by up to about 20 °C in a day and by 40 - 50 °C in a year. Existing studies suggest that the absolute response of the instruments may change by 0.2 – 0.3% per 1 °C change of the internal temperature (Garane et al., 2006; Lakkala et al., 2008). Thus, not accounting for temperature effects may lead to uncertainties or biases greater than the desirable overall uncertainty in the measurements.

In contrast to the correction of TOC measurements for the effects of temperature, which is achieved using a standard methodology (Kipp & Zonen, 2008; Savastiouk, 2005; SCI-TEC Instruments Inc., 1999), there is no standard method for the characterization or the correction of the Brewer global UV measurements. At several sites, 50, 200 and 1000 Watt tungsten – halogen lamps which are used to monitor the stability and/or calibrate the Brewer spectrophotometers (e.g. Bais et al., 1996; Bernhard et al., 2008; Heikkilä et al., 2016) are also used for the temperature characterization of the instruments. These lamps are usually warm-colored, with a color temperature of ∼3000K, thus they emit a significant amount of infrared radiation. They are usually placed at vertical distances of 5, ∼20 and 50 cm from the diffuser, respectively, with the center of their filament aligned with the center of the diffuser. Measurements are performed either inside, or outdoors at the regular operating position of the instrument, using various setups which may differ between individual stations. In Cappellani and Kochler (2000),Siani et al. (2003) and Weatherhead et al. (2001) the temperature characterization of the global UV measurements was achieved by performing measurements of the irradiance from 50 Watt lamps outdoors, at different ambient temperatures. In two more recent studies (Garane et al., 2006; Lakkala et al., 2008) the reported temperature correction factors were derived by performing measurements with 1000 Watt lamps in the laboratory. Using the internal 20 Watt standard lamp (sl) (Kipp & Zonen, 2008) for temperature characterization may lead to errors since the emissivity of the lamp may be affected by the temperature effects on the lamp's power supply (Weatherhead et al., 2001). Furthermore, when the sl is on, it is possible that large temperature gradients exist inside the instrument. In addition, any changes in the transmittance of the diffuser are not taken into account when the internal lamp is used. All the above studies have reported that the Brewer's response is a monotonic linear function of temperature which may also depend on wavelength. However, Ylianttila and Schreder (2005) have found a sharp change in the transmittance of the Poly-tetrafluoroethylene (commonly known as Teflon) diffusers near 19°C, usually ranging from 1% to 3%, which should also affect the response of Brewer spectrophotometers. This temperature behavior has not been confirmed so far for the diffusers used in Brewer spectrophotometers.

In the present study we investigate the effects on Brewer diffusers when 50, 200 and 1000 Watt tungsten – halogen lamps are used for the characterization of the temperature dependence of spectral UV irradiance measurements. Additionally, the results of the characterization for the effects of temperature, of the single monochromator Brewer spectrophotometers with serial numbers 005, 030, 037, 078 (now on referred as B005, B030, B037, B078 respectively) and the double monochromator Brewer spectrophotometers with serial numbers 086, 107, 185 and 214 (now on referred as B086, B107, B185 and B214

respectively), are analyzed and compared to each other. These instruments are operating regularly at Thessaloniki, Greece (B005 and B086), Sodankylä, Finland (B037 and B214), Helsinki, Finland (B107), Lindenberg, Germany (B030 and B078) and Izaña, Tenerife, Spain (B185). Finally, the application of the derived temperature correction factors is evaluated for the Brewer spectrophotometers operating at Thessaloniki and Sodankylä.

## 2 Evaluation of the temperature characterization with external lamps

As listed in the Introduction, the temperature dependence can be due to different reasons. In this study, we investigated how the error due to the temperature dependence can be corrected using the only available information, which a standard Brewer user has: the PMT temperature. We study which method is best to characterize the Brewer (50 W, 200 W, 1000 W lamps) and which are the uncertainties related to this method. For this purpose measurements with 50 and 1000 Watt lamps were performed by B005, B086 and B214 inside the laboratory, while measurements with 200 Watt lamps were performed by B185 at the site of the instrument's regular operation.

Characterization of B214 with 50 and 1000 Watt lamps was performed using the methodology described in Lakkala et al. (2008). Measurements with both types of lamps were performed under the same conditions, so that any differences in the results could be attributed to effects of the used lamps. The change in the response of B214 at various temperatures relative to 25°C for 330 nm is presented in Fig. 1. While the results from the 50 Watt lamp indicate that the response of the Brewer decreases linearly with temperature, the 1000 Watt lamp reveals a sudden increase in the response for temperatures between ~13 and 22°C, and for higher and lower temperatures the pattern is similar with the 50 Watt lamp. Similar results are obtained for all wavelengths. The characterization of B005 and B086 with 50 and 1000 Watt lamps yielded patterns which were also similar with those presented in Fig. 1. It should be noted that the characterization procedure followed for these two instruments (Garane et al., 2006) is slightly different compared to that of Lakkala et al. (2008).

The different behavior of the Brewer's temperature dependence could be attributed only to the different thermal effect of the two lamps on the diffuser, taking into account that although weaker, the 50 Watt lamps are placed much closer to the diffuser. In order to investigate further this effect, the temperature of the diffuser of the two Brewers operating at Thessaloniki was measured with a VOLTCRAFT IR 260-85 infrared thermometer while each lamp was turned on. The lamps were positioned at the same distances and with the same configuration as in the absolute calibration and the temperature characterization tests. Initially, only the cover of the Brewer that holds the diffuser with the dome was used and the temperature was measured by pointing the IR thermometer towards the center of the diffuser from beneath, i.e. from the side where normally the fore optics of the Brewer are located. The temperature of the diffuser was measured when it was not illuminated by the lamp. The lamp was either moved away (50 Watt lamps) or the radiation was blocked (1000 Watt lamps) before each measurement. This procedure was repeated several times for about 90 mins. During the test the ambient temperature in the vicinity of the diffuser was also monitored. The temperature of the diffuser and the ambient as a function of time are presented for B005 in Fig. 2a and 2b, respectively for the 50 Watt and the 1000 Watt lamps. The behavior of B086 is almost identical to that of B005.

For the 50 Watt lamp (Fig. 2a) the temperature of the diffuser increases fast, by about 20 °C in the first 30 mins, and thereafter it remains relatively stable, while the ambient temperature remains almost stable. Moreover, measurements at different parts of the diffuser's surface while it was illuminated revealed inhomogeneities of 5 – 6 °C. These findings suggest that 50 Watt tungsten-halogen lamps are not suitable to characterize the overall temperature dependence of the Brewer, since they affect the temperature and eventually the throughput of the diffuser.

For the 1000 Watt lamp, the temperature of the diffuser increases gradually by 5 – 8 °C following the almost identical increase of the ambient temperature in the dark room (Fig. 2b). This suggests that the lamp does not affect significantly the temperature of the diffuser, or at least no more than it affects the ambient temperature.

For the 1000 Watt lamps we tested also an alternative configuration: The whole instrument (not only the cover) was placed under the lamp and the temperature of the diffuser was recorded by pointing the IR thermometer towards the upper surface of the diffuser after removing temporarily the quartz dome. During each measurement the radiation of the lamp was blocked. Then the dome was restored to its position and the lamp was unblocked to illuminate the diffuser until the next measurement. Each measurement (from blocking to unblocking the light) did not last more than ∼1 min. During this test, in addition to the ambient temperature of the dark room, the temperature at the PMT was also recorded using a built-in thermistor. The measurements collected from this test are presented in Fig. 2c. As with the cover only, the temperature of the diffuser is not affected by the lamp's radiation and increases gradually following the increase of the temperature at the PMT. This suggests that the temperature at the PMT could represent both the internal temperature of the Brewer and that of the diffuser for the characterization procedure. However, during regular outdoors operation the temperature of the diffuser may differ significantly from the temperature of the PMT, as discussed later.

The slower increase of the diffuser's temperature compared to Fig. 2b is possibly explained by the fact that the lower surface of the diffuser is not exposed to ambient air, protected by the cover and the body of the instrument. Finally, it may be concluded that the 1000 Watt lamps, or any type of lamps that are positioned adequately far from the diffuser to prevent direct heating, can be safely used to characterize the temperature response of the Brewer.

At the station of Izaña, the temperature of the diffuser of B185 was recorded using an infrared sensor while a 200 Watt lamp was placed above the diffuser. In this case the sensor was adjusted inside the instrument, aiming at the bottom surface of the diffuser and its signal was recorded by a data logger which was also placed inside the instrument. The setup of the 200 Watt lamp is the same as that used for the regular monitoring of the instrument's stability and for the temperature characterization. The measurements were performed at the location of regular operation of B185. As for the Brewers in Thessaloniki, no significant change of the temperature of the diffuser was detected while it was illuminated by the 200 Watt lamp.

The lamps used at other sites usually have similar characteristics with those used at Thessaloniki, Sodankylä, and Izaña, and even if they are not supplied from the same manufacturer they are expected to have similar effects on the temperature of the diffuser.

## 3 Characterization for the effects of temperature

The results presented in this section were obtained by characterizing the temperature sensitivity of the instruments' response in the laboratory. Several instruments were tested following slightly different procedures according to facilities available in each station. The temperature of the PMT which is regularly monitored has been used to derive the temperature correction factors. In Thessaloniki, the B005 and B086 were moved from the site of the instruments' regular operation inside the calibration room during a cold day and irradiance measurements of a 1000 Watt lamp were performed as the room temperature and the internal temperature of the instruments were gradually increasing. The increase in temperature was slow enough to ensure that the temperatures of the room and the instrument are equilibrated (Garane et al., 2006). In Sodankylä and Helsinki, B037, B107 and B214 were also carried inside the laboratory, where irradiance measurements were performed similarly as during a calibration (Lakkala et al., 2008), but the temperature of the Brewer was slowly increased or decreased using an air-blower system built for the specific purpose (Lakkala et al., 2008). Before each measurement, the temperature was stabilized and remained constant during the measurement. In Lindenberg, B030 and B078 were placed in a chamber wherein the temperature was increased or decreased slowly and scans were performed using a 200 Watt lamp after the temperatures outside and inside the instrument were fully stabilized and temperature gradients were practically zero. B185 was also placed in a chamber at the facilities of PTB and the temperature characterization was obtained similarly as for B030 and B078 (Berjón et al., 2016). In all the above cases the current of the lamps was constant within less than 1 mA (8 A for the 1000 Watt lamps and 6.3 A for the 200 Watt lamps) during measurements. The spectrum of the used lamp was measured before and after the characterization of B037, B107 and B214 to ensure that neither the response of the instruments nor the characteristics of the lamp changed during the characterization procedure. For the remaining five Brewers the signal of the lamp was recorded using either a photodiode (B005, B086, B030 and B078) or a silicon detector and a CCD spectrometer (B185) to ensure that the detected changes are not due to changes of the lamp's emission. In all cases a line (the 297 or 302.15 nm line depending on each instrument settings) of the internal Hg lamp (Kipp & Zonen, 2008) was scanned before measurements at specific temperature to ensure wavelength stability. All measured spectra were corrected for the effects of the dark signal and the dead time, and smoothed by a 3-point moving average filter to suppress the noise. Finally, the change of the instrument's response with respect to temperature was calculated.

Brief information regarding the site of regular operation and the characteristics (type, single or double monochromator, NiSO4 filter or not) of each instrument is summarized in Table 1. A short description of the place and the method of characterization and the temperature range are also provided in the last two columns.

### 3.1 Temperature correction factors for different Brewers

Analysis of the measurements of the eight Brewer spectrophotometers revealed three temperature ranges (TR) with different patterns in the temperature response: Low (TR1), around 19 °C (TR2) and high (TR3). In these ranges spectral measurements of the global UV irradiance should be treated differently in order to correct for temperature effects. In each range the temperature dependence of the instrument's response can be described to a good approximation by least-squares linear fit. Although for each

individual instrument the slopes of the linear fits change with wavelength, the limits of the temperature ranges are wavelength independent. Indicative results for 315 nm and for all the eight instruments are presented in Fig. 3. The limits were estimated by eye and then the linear fit that describes the change in response for each TR was calculated.

Measurements below 10 °C, which provide information for TR1, were possible only for three of the instruments (specifically for B078, B185 and B214). At this point it should be noted that Brewer spectrophotometers are equipped with a heater which is automatically turned on when the internal temperature drops below a specific limit, either 10 °C or 20 °C (Kipp & Zonen, 2008). Thus, even for ambient temperatures below zero, the internal temperature does not usually drop below 0 °C and 10 °C respectively. For the same reason it is difficult to perform measurements for such low internal temperatures during the characterization procedure. For B185 and B214 the heater threshold has been set to 20 °C while for the other six instruments it is set to 10 °C. To achieve low internal temperatures for the first two the heaters were disconnected during the characterization procedure. For B185 two sets of measurements were performed: one for internal temperatures ranging from -2 to 24 °C and one for temperatures between 13 and 50 °C. The measurements of the two sets were analyzed independently. Prior to analysis, the measurements were normalized to the highest common temperature of the two sets (24 °C). For all the other instruments the measurements were normalized to the 20 – 30 °C means.

The results presented in Fig. 3 verify that the response changes differently in the three TRs. However, the limits of the different TRs were not found to be the same for all instruments and even for the same instrument they may differ if the characterization is repeated under different conditions. For example, for the two sets of B185 TR2 was respectively 13 and 9 °C, i.e. about 4 °C different. For B078 and B214 the TR2 was 6 and 10 °C respectively. For the remaining five instruments, for which it was not possible to determine clearly the boundaries of TR2, this was assumed to be 10 °C. This approximation was made since this range was found close to ∼10 °C for B078, B185 and B214. The limit that separates TR1 from TR2 ranges between 12°C (for B214) and 16°C (for B078), while the limit that separates TR2 from TR3 ranges from 20°C (for B078) to 26°C (second set of B185). The differences between the TR limits and ranges which were found for each case are estimated to be mostly related with the uncertainties in the characterization results and less with the characteristics of each instrument.

For B078, B185 and B214 the differences between the slopes of the linear fits for TR1 and TR3 are small and although the slopes for TR1 were derived either from a limited number of measurements (B078, B214), or with the internal heater turned off, which does not represent realistic operational conditions (B185, B214), the results provide a strong indication that the slope calculated for TR3 can be also used for the correction of measurements within TR1, without inducing important errors. It is noteworthy that for seven of the eight studied cases the response is increasing in TR2 and decreasing in TR3, while increase in both, TR2 and TR3, was found only for the case of B005.

In Fig. 4, the calculated slopes are presented as a function of wavelength for TR2 and TR3. For B185 only the results from the second set of measurements (for higher temperatures) are presented. For all cases the dependence from wavelength is described satisfactorily by a 2nd degree polynomial. With the exception of B185 (TR2 and TR3) and B030 (TR2), for all the other instruments the percentage change of their response for 1°C increase in temperature increases with wavelength. The dependence of the slope from wavelength is stronger for the single compared to the double monochromator Brewers, possibly due to the presence of the $UG11 \smile NiSO_4$ filter combination at the entrance of the PMT of the former. The change of the

response in TR2 ranges from 0%/°C (for B078) to 0.6%/°C (for B005). In TR3 the change of the response ranges from – 0.3 to +0.2% for different instruments and wavelengths.

Analytical description of the methodology that should be used for the correction of the global UV irradiance measurements, as well as the calculated correction factors for each of the eight Brewers can be found in the supplement.

## 3.2    The role of the diffuser

Comparison between the patterns shown in Fig. 3 for all Brewers (i.e. decreasing, or slowly increasing, response as temperature increases for TR1, fast increasing response for TR2, and again decreasing, or slowly increasing, response for TR3), with the results of Ylianttila and Schreder (2005) leads to the conclusion that part of the observed changes is due to the effect of temperature on the transmittance of the diffuser. However, the slope of the linear fit in TR1 and TR3 is in most cases different than what would be expected according to their results indicating that part of the dependence also is due to the effect of temperature on the PMT and other internal optical and mechanical components of the instrument.

To investigate the validity of this assumption, spectral irradiance measurements were performed at different temperatures using an external lamp through the slant quartz window, and the internal standard lamp of the Brewer. The results were then compared with those from the measurements through the diffuser. These measurements were performed by B005, B086 and B185 using slightly different setups. For all the three Brewers it was found that, while for the measurements through the diffuser the response changes differently in TR1, TR2, and TR3, for the measurements of the external lamp through the window and the sl the response changes with the same rate in the entire range of recorded temperatures. The results for B005 are presented in Fig. 5.

In Fig. 5a, the linear fits of the results are also presented. In Fig. 5b the slopes of the least square linear fits and the corresponding $1\sigma$ uncertainty in their determination are presented, as well as the second degree polynomials which describe the dependence from wavelength in each case. These results confirm that the different patterns found between the three TRs are mainly due to the change in the transmissivity of the Teflon diffuser. For the measurements through the window it was found that the change in the response/°C is wavelength dependent for both the single and the double monochromator Brewers, indicating that the dependence of wavelength might not be introduced solely by the $NiSO_4$ filter used only in the single monochromator Brewers, as suggested by Weatherhead et al. (2001) and Garane et al. (2006). Some possible explanations for the different results for the sl and the measurements through the window are the following: (a) large temperature gradients exist inside the instrument when the sl is on, (b) the electronic circuits of the sl may be affected by temperature and (c) the transmittance of the quartz window may be affected by temperature. However, further investigation is out of the scope of the present study. In order to clarify whether the changes of temperature may also affect the transmittance of the quartz dome, spectral irradiance measurements with and without the dome were performed by B086 and the mean spectral transmittance of the dome was derived for different temperatures. It was found that for temperatures ranging between ∼15 and 45°C the dome blocks ∼6% of the incoming radiation, independently of temperature or wavelength.

Ylianttila and Schreder (2005) measured the transmissivity of a number of radiometric instruments, none of which was a Brewer, and found that the effect of temperature on the transmissivity of the Teflon diffusers mainly depends on their thickness

and the wavelength of the incident irradiance. For the cases they studied, they found that near 19°C the temperature transmissivity changes range between ~1 and 3% which is in good agreement with our results. The width of TR2 also seems to differ by a few °C between the different instruments used in their study. The differences between the changes of response and the width of TR2 (even when the same characterization methodology was used) for the Brewers used in this study denote that the individual characteristics of each diffuser may play an important role on its correspondence to the changes of temperature. However, part of these differences is also related to the uncertainties in the characterization procedure as explained in the following.

## 3.3 Uncertainties in the characterization and the correction

Ideally, the characterization should be performed separately for the effect of temperature on the transmittance of the Teflon diffuser, the transmittance of the internal optical and mechanical components and the response of the PMT. However this is not possible due to insufficient information to partition the effect among the different components, as well as due to lack of systematic recording of temperature at each component. A parameterization including the PMT and the ambient (environmental) temperature might also describe more accurately (than using only the PMT temperature) the effect of temperature since this way the possible differences between the temperature of the PMT and the diffuser would be partially taken into account. However, this would make the characterization very complicated. Furthermore, the environmental temperature out on the sun, where the Brewers are routinely operating, is different than the temperature provided by the meteorological stations which is measured in the shadow and the former is not usually recorded. Therefore, the characterization and correction is performed for the overall response to temperature using the temperature that is recorded by the thermistor attached to the PMT for each single spectral scan. The assumption that this temperature is representative for all parts of the instrument introduces some uncertainties, which are difficult to quantify.

### 3.3.1 Temperature gradients inside the instrument

During the characterization procedure, the 200 and 1000 Watt lamps do not warm the diffuser, and as long as the warming or cooling of the Brewer is slow the differences between the temperature of the diffuser and the PMT can be considered negligible. However, during the regular operation of the Brewer, larger differences may exist. A suitably designed infrared sensor was installed inside B185 to record the temperature of the lower surface of the diffuser for about a month during which the instrument was operating regularly outside and the ambient temperature was ranging between $sim$ -2 and 27 °C. Analysis of the results revealed that when the internal heater is off, the difference is generally smaller than 4°C showing that even large differences between the environmental and the internal temperature, which are not unusual for Brewers (Weatherhead et al., 2001), do not imply correspondingly large differences between the temperatures of the PMT and the diffuser. However, when the heater is on (below 20 °C), temperature gradients appear in the instrument, which become more important as the temperature decreases. Under these conditions, the temperature of the diffuser is much lower than the temperature of the PMT, and for the lowest recorded internal temperatures (15 – 16 °C) differences of up to 10 °C can be encountered. For instruments for which the heater is turned on at 10 °C, the gradients are expected to be important at lower temperatures. Based on the

results of Ylianttila and Schreder (2005) and the results presented in Fig. 3 we estimate that the errors in the correction of measurements due to the difference between the recorded and the actual temperature of the diffuser when the heater is on are, in all the studied cases, smaller than 2%.

### 3.3.2 Hysterisis of the PMT

5 The hysteresis is not solely related to the temperature gradients inside the instrument when the heater is on. The interior of the PMT is a vacuum and heat conducts through it very slowly. Thus, the PMT reaches the temperature level of the environment later than other parts of the instrument and it is questionable whether the recordings of the thermistor which is attached to the PMT housing represents its actual temperature or the temperature of the housing. Hysteresis loops that have been also observed when measurements of the standard lamp were analyzed with respect to temperature (for B005, B086 and B185), as well as 10 analysis of the characterization results for high temperatures (for which the heater is turned off) confirm this assumption. Though, the hysteresis due to the delay in the response of the PMT is estimated to have a minor impact on the overall behavior of the instrument compared to the impact of the differences between the temperature of the diffuser and the PMT.

### 3.3.3 Effect of temperature on the determination of the spectral response

The determination of the spectral response of the Brewers is usually performed in the laboratory using 1000 Watt lamps. As 15 shown in Fig. 2, the lamps warm the air in the calibration room, which leads to a gradual increase of the instrument's internal temperature. Depending on the instrument and the measurement settings determined by the operator, each scan of the lamp's spectrum may last from a few ($\sim$3 – 5) to several ($\sim$20-30) mins. Thus, according to the results presented in Fig. 2, the temperature of the instrument changes during each spectral scan and at the end of the scan it may differ by a few °C. In the case of Thessaloniki the calibration room is small (a few m$^2$) and the lamp warms the air in the room fast. Performing the 20 calibration in a bigger room and/or improving the ventilation would lead to slower changes of the temperature. Obviously, the calibration factor should be derived for a standard temperature, or alternatively all measurements should be interpolated to the temperature of the calibration. For this purpose, the temperature which is recorded at the beginning of each scan of the lamp can be used. Based on the results presented in Figs 2 and 3 we estimate that the uncertainty in the determination of the calibration factors due to the changes of temperature during the calibration procedure will be less than $\sim$0.5%, given that in most cases 25 the calibration is performed at temperatures near 25 °C or higher, and the change of temperature during each spectral scan is less than $\sim$5 °C.

### 3.3.4 Heating and cooling rate during characterization procedure

If the rate of heating or cooling during the characterization procedure is not slow enough, non-negligible temperature gradients may appear inside the instruments. Thus slightly different heating or cooling rates during the characterization procedure 30 may lead to the calculation of slightly different correction factors. This may explain the differences between the two sets of measurements with B185 as well as the large spread in the measurements of B030 and B078 shown in Fig. 3.

### 3.3.5 Photon noise

When the signal of the lamp is low, the uncertainty in the measurements, and consequently in the characterization results, due to the photon noise may be also important (e.g. Grajnar et al., 2008). In these cases, increase of the exposure time of the PMT may improve the results. For example, the measurements for the characterization of B005 were initially performed with an exposure time of ∼0.45 sec (the results are not presented in this study) and then were repeated with an exposure time of ∼4.5 sec (results presented in Figs 3 and 4). Although the number of data points was similar in the two cases, the standard deviation in the correction factors was ∼10 times larger when the exposure time was smaller. These uncertainties are generally more important at lower wavelengths where the signal of the lamp is weaker.

### 3.3.6 Effect of temperature on the wavelength stability

One more possible factor of uncertainty in the characterization procedure is the apparent responsivity change due to the effect of temperature on the wavelength stability of the instrument. Temperature changes lead to change of the instrument's spectral characteristics (Gröbner et al., 1998). To compensate for this effect the 297 (or 302) nm line of the internal Hg lamp is scanned when the temperature changes by ∼1-2 °C and the zero position of the micrometer is adjusted properly (Grajnar et al., 2008). However, the correction based on the results for 297 or 302 nm does not ensure that there are no wavelength shifts at larger wavelengths. Spectra of the global solar irradiance measured by B005 and B086 were analyzed using the SHICrivm algorithm (Slaper et al., 1995) and no significant dependence of the wavelength shift from temperature was found. However, even if we assume a small shift in wavelength (i.e. of 0.01 nm) during the characterization procedure, this would induce an important apparent responsivity change (of the order of 1%) only for the single monochromator Brewers near 325 nm where the spectral response of the instrument changes fast.

### 3.3.7 Number of spectra during characterization

The low number of measurements also increases the uncertainty in the characterization results, especially when the recorded signal and the exposure time of the PMT are also low. The finite, usually low number of measurements in the TR1 and TR2 induce uncertainties in the determination of the TR limits and the correction factors. Thus, for each instrument, slightly different TR1 - TR2 and TR2 - TR3 limits may be found when the characterization is repeated. Analysis of the characterization results for four different sets of measurements (performed in different days of 2005) with B086 resulted in TR2 - TR3 limits ranging between 24 and 28 °C. The same analysis for B005 resulted in smaller differences (22 – 24 °C). Separate analysis of the results for the warming and the cooling of B185, for the second set of measurements, also lead to different TR2 - TR3 limits, at ∼30 °C and ∼23 °C respectively. We estimate that the uncertainties in the correction factors due to these differences cannot exceed ±0.5%.

## 3.4 Evaluation of the derived correction factors

For the evaluation of the results presented in Sect. 3.1, global UV spectra that were measured nearly simultaneously by the two Brewer spectrophotometers operating at Thessaloniki (B005 and B086) and by those operating at Sodankylä (B037 and B214) were compared to each other. For Thessaloniki, measurements for 15 years (2001 – 2014) were used in the comparison while for Sodankylä measurements were available only for a period of six months (April – October 2016). More specifically, the 300 – 325 nm integrals of spectra measured within 1 minute were compared for each pair of collocated instruments, before and after applying the temperature correction. Since changes in temperature affect the measurements of each instrument differently, it is expected that the ratio of the uncorrected for the effect of temperature data between two instruments will be temperature dependent, and that the greatest part of this dependence would be eliminated when temperature corrected data are used instead. These ratios normalized to the mean ratio at 25 °C are shown in Fig. 6 for Thessaloniki and Sodankylä Brewers as a function of temperature recorded, respectively, by B086 and B037. The error bars represent the 1-sigma standard deviation of the mean for each 10 °C bin.

According to Fig. 6a, the temperature correction of the data of B005 and B086 almost eliminates the otherwise strong temperature dependence of their ratio. Similar results are achieved for the two Brewers operating in Sodankylä. Despite the lower temperatures in Sodankylä which may result in large temperature gradients inside each instrument when the heaters are turned on, the results verify that the applied correction is towards the right direction, even for ambient temperatures of about -10 °C.

Temperature correction factors were determined for B005 and B086 operating in Thessaloniki twice, in 2005 (Garane et al., 2006) and 2015. The differences in the derived temperature correction factors are smaller than their $1\sigma$ uncertainty suggesting that these correction factors are valid for the entire period (2001 – 2015). This has been also confirmed from the comparison of quasi-simultaneous measurements of the two instruments for this period. In contrast, the comparison of measurements for the period 1993 – 2000 revealed that the correction factors cannot remove effectively the dependence effects for this period. This can be attributed to the replacement of the PMT of B086 in 2000, which has different temperature response compared to the old PMT. Replacement of other electronic parts of B005 and B086 was not found to induce detectable changes in their behavior regarding the effects of temperature. The above indicate that the temperature dependence does not change significantly with time as long as the components of the instrument that are mainly affected by changes in temperature (i.e. the Teflon diffuser and the PMT) remain the same.

## 4 Conclusions

The sensitivity of the Brewer spectrophotometers in spectral irradiance measurements shows a marked dependence to temperature variations. Thus, the use of uncorrected spectra for the study of the diurnal, seasonal and annual changes of UV irradiance would lead to inaccurate results due to the corresponding cycles of temperature. Although improper correction of the spectra for the effects of temperature would not possibly have an important effect on the study of the long term changes of the UV irradiance at low and mid-latitudes, it may be more important for higher latitudes where the annual mean temperature is chang-

ing, and is projected to keep changing fast in the following decades (IPCC, 2007). The Accurate correction of the spectra for the effects of temperature would improve the agreement between the measurements from different Brewers and lead to a more reliable product which in turn would be suitable for climatological studies and the validation of satellite products and model simulations.

The % difference of the 315nm response from its value at 25 °C due to the effect of temperature is presented in Fig. 7 (left column). The presented results are for all the spectra measured in 2016 and for all the eight Brewers used in this study. The corresponding temperatures recorded at the PMT are presented in the right column of the same figure. The differences from the response at 25 °C have been calculated using the results of Sect. 3.1.

Depending on the site and the instrument, the response may differ by up to 6% (e.g. in the case of B005) in a year. These
differences seem to be smaller (occasionally of the order of 1.5% in a day) for instruments which operate at low temperatures and the threshold of the heater is 20 °C (e.g. B185 and B214). However, as explained earlier, the correction of the measurements from these instruments is more uncertain due to the large temperature gradients when the heater is on. According to the results presented in Fig. 7, there are upper and/or lower limits in the changes of the response for most Brewers. These limits correspond to the turning points (limits between TR1 and TR2 and between TR2 and TR3) of the lines presented in Fig. 3 and are evident
because for all studied cases the temperature ranges between a lower (between 0 and 20 °C) and an upper (between 30 and 50 °C) limit.

The temperature response of Brewers, either these are single or double monochromators, is wavelength dependent and instrument specific. The main components of the Brewer spectrophotometers that determine their behavior relative to temperature variations are the PMT, the diffuser and possibly the $UG11 - NiSO_4$ filter in the single monochromator Brewers.

For irradiance measurements through the Teflon diffuser (global irradiance) the response is not unique for the entire range of operating temperatures, mainly because of the steep increase (decrease) in the transmissivity of the diffuser as the temperature increases (decreases) from ∼12 (22) to 22 (12) °C. It is suggested that different correction factors should be used for three different temperature ranges (TR1: below ∼12 °C, TR2: ∼12 – 22 °C, TR3: above ∼22 °C). The temperature dependence is very similar in TR1 and TR3, thus applying the same correction factors for these ranges does not introduce large uncertainties.

Characterization for the effects of temperature using the 50 Watt lamps, which are operationally used to monitor the stability of the Brewer spectrophotometers, may lead to wrong correction factors since these lamps are positioned very close to the diffuser, increase its temperature and alter its transmissivity. The 1000 Watt lamps, regularly used for the absolute irradiance calibration at distances longer than 50 cm, do not heat the diffuser and lead to more reliable results when they are used for the temperature characterization. The setup with 200 Watt lamps, which is used at several stations for monitoring the instruments'
stability, is also suitable for temperature characterization because the distance of the lamp from the diffuser is adequately long to prevent direct heating.

The proposed methodology which is described in detail in the supplement was evaluated using spectra from the Brewer spectrophotometers operating in Thessaloniki and Sodankylä and was found to remove the greatest part of the temperature dependence from the irradiance measurements. The correction of the spectra using the specific methodology is more accurate
compared to the correction based on the methodologies described in previous studies (Garane et al., 2006; Lakkala et al., 2008;

Siani et al., 2003; Weatherhead et al., 2001) since the effect of temperature on the transmissivity of the diffuser is also taken into account. The correction factors for each Brewer depend on its individual constructional characteristics, thus it is not possible to apply generic correction for all Brewer spectrophotometers and characterization of each individual instrument is necessary. Repeating the characterization procedure frequently was not found necessary as long as the main components of the instrument

which are affected by temperature variations are not replaced.

The uncertainties in the calculated correction factors are small as long as the warming and cooling of the instrument is slow enough to prevent the development of large temperature gradients inside the instrument during the characterization procedure. Increasing the number of measured spectra and/or the exposure time of the PMT, especially at temperatures between ∼10 and 25 °C, may lead to smaller uncertainties in the derived correction factors. Uncertainties in the correction of global irradiance

spectra arise mainly from the use of the temperature recorded at the PMT to correct the measurements. Large temperature gradients inside the instrument when the heater is turned on, may occasionally lead to large differences, of the order of 10 °C, between the actual temperatures of the PMT and the diffuser. For particular instruments, these differences may subsequently lead to errors of up to 2% in the correction of spectra recorded with the heater on. The uncertainties due to the characterization procedure and the methodology to derive the correction factors are estimated to be generally smaller than 0.5% and more

important for temperatures below ∼25 °C.

*Competing interests.* No competing interests are present

*Acknowledgements.* This article is based upon work from COST Action ES1207 "A European Brewer Network (EUBREWNET)", supported by COST (European Cooperation in Science and Technology) and from the ENV59-ATMOZ ("Traceability for atmospheric total column ozone") Joint Research Programme (JRP). The JRP is jointly funded by the EMRP participating countries within EURAMET and the

European Union. All the members of the COST Action ES1207 who contributed, either by providing data which were finally not used in the present study or through constructive discussions are also acknowledged. Special thanks to J. Gröbner for his helpful advices. The anonymous reviewers are also acknowledged for their constructive reviews that helped improving the quality of this paper.

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

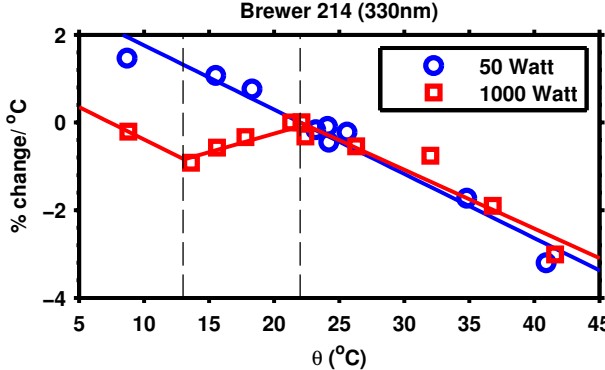

**Figure 1.** Changes in the response of B214 at 330 nm, relative to the response at 25 °C for the same wavelength, as a result of changes in temperature, derived using 50 and 1000 Watt tungsten – halogen lamps.

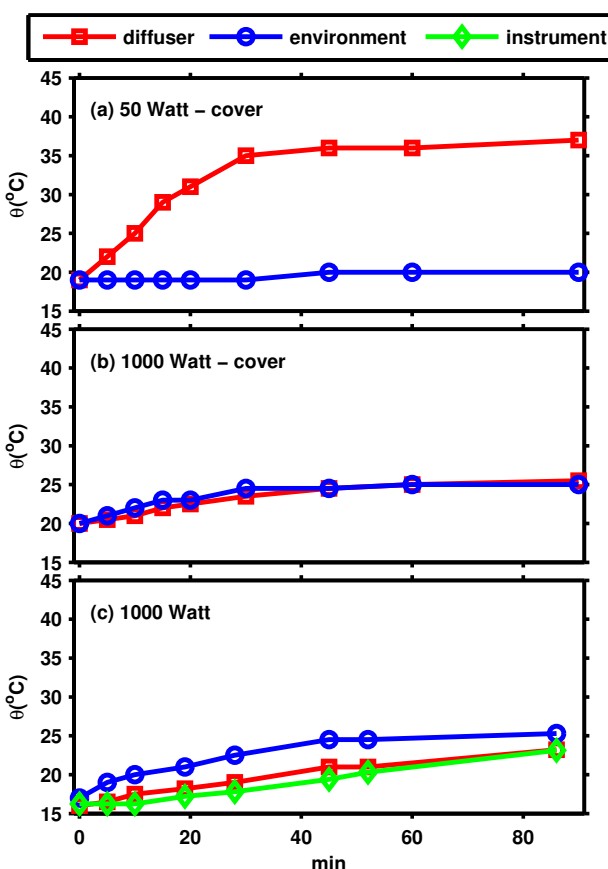

**Figure 2.** Change of the temperature of the diffuser of B005 (a) when a 50 Watt lamp is placed at a vertical distance of 5 cm above the diffuser and only the cover is used, (b) when a 1000 Watt lamp is placed at a vertical distance of 50 cm above the diffuser and only the cover is used, (c) when a 1000 Watt lamp is placed at a vertical distance of 50 cm above the diffuser and the instrument is operating normally (the instrument is inside the cover).

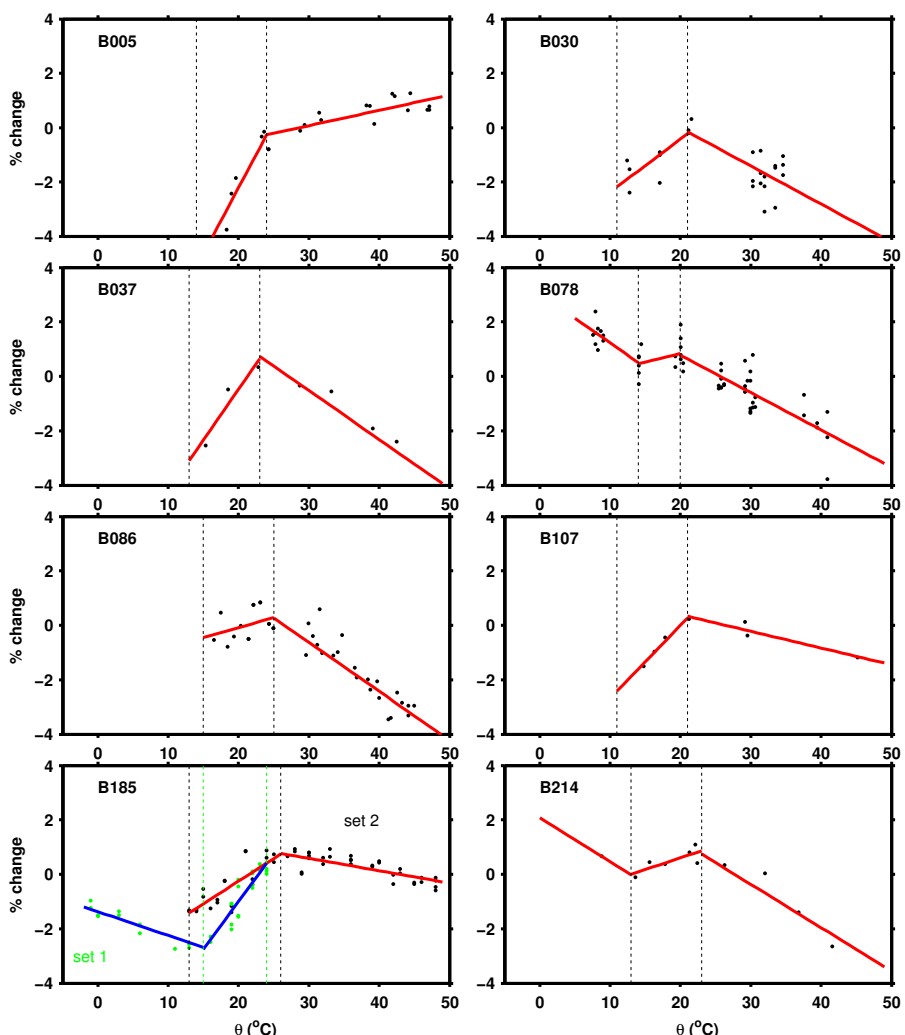

**Figure 3.** Change (in %) in irradiance at 315 nm with respect to the instrument's internal temperature for eight Brewer spectrophotometers. The estimated limits that separate the three TRs for each instrument are represented by the two dotted lines, while the linear fits that describe the change in each instrument's response are represented by the red lines. For B185 the two sets of measurements are represented by different colors (green and black) as well as the corresponding linear fits (blue and red).

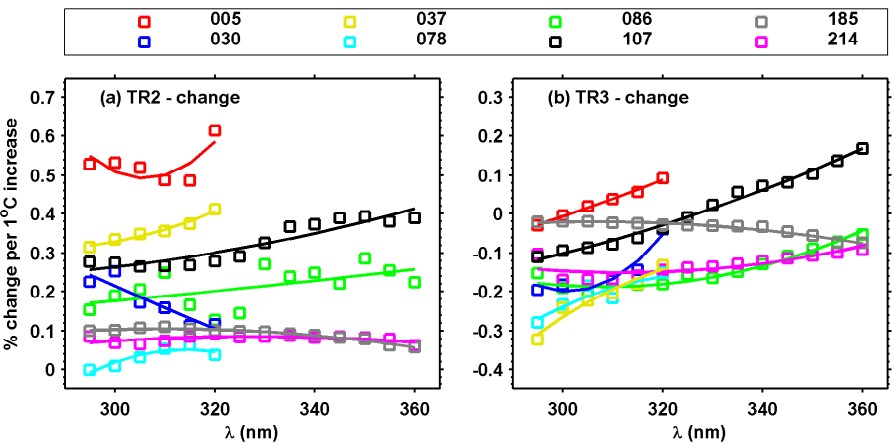

**Figure 4.** Change (in %) of irradiance for 1°C increase in temperature as a function of wavelength for (a) TR2 and (b) TR3 for the eight Brewers.

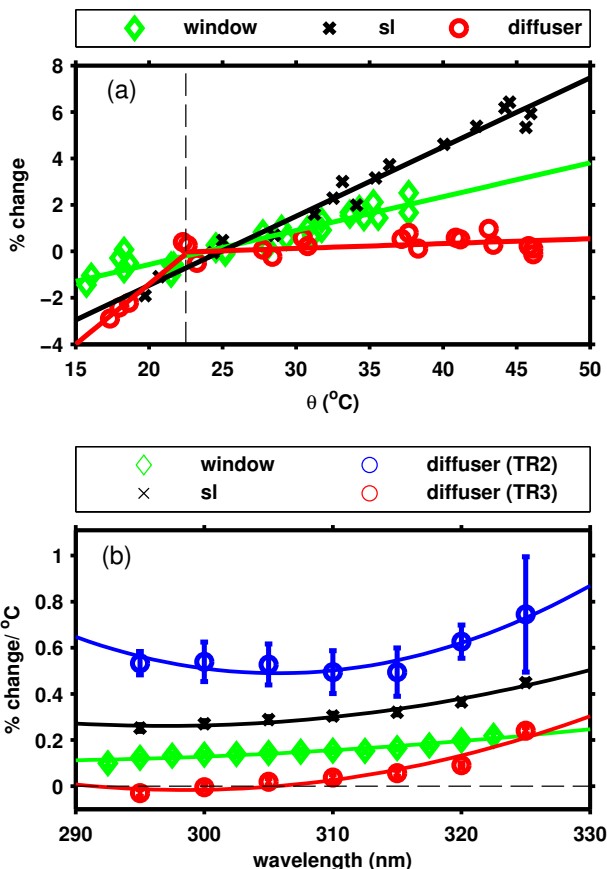

**Figure 5.** (a) Change in the response of B005 as a function of temperature for the sl and measurements through the window and the diffuser, relative to the response at 25°C and (b) the dependence of the derived slopes from wavelength.

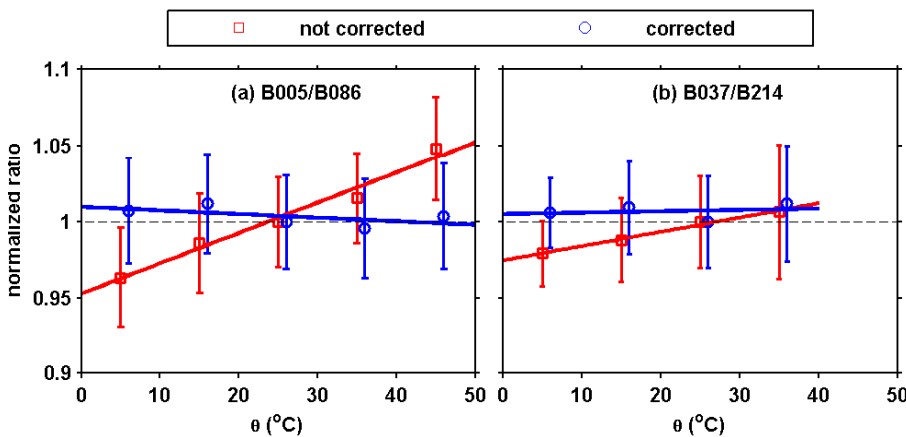

**Figure 6.** Ratio of the 300 – 325 nm irradiance integrals derived for each pair of Brewers as a function of temperature before and after applying a temperature correction for (a) B005 and B086 in Thessaloniki and (b) B037 and B214 in Sodankylä. The ratios have been normalized to the mean ratio at 25 °C. The error bars represent the 1-sigma standard deviation of the mean for each 10°C bin.

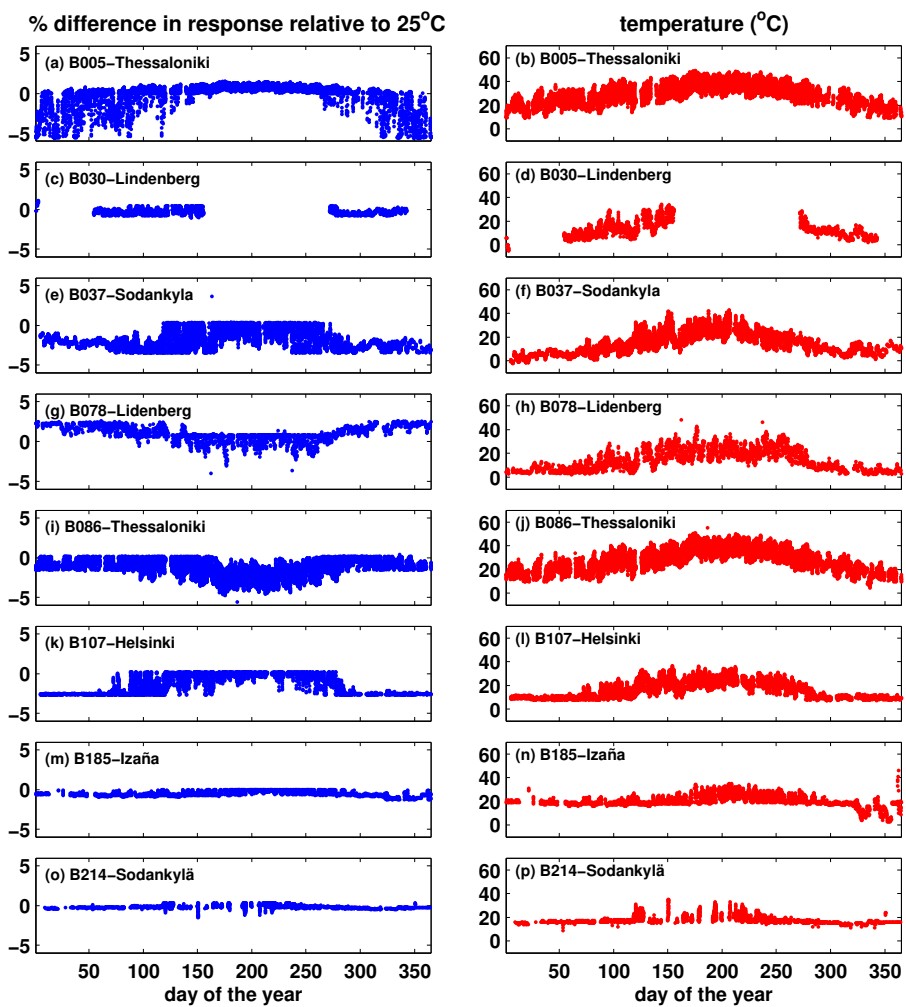

**Figure 7.** Differences (%) of the 315nm response from its value at 25 °C due to the effect of temperature and the corresponding temperatures.

**Table 1.** Information for each Brewer spectrophotometer and the corresponding temperature characterization procedure

| Site | Instrument (Type) | Monochromator ($NiSO_4$ filter) | Characterization Method | Temperature range (°C) |
|---|---|---|---|---|
| Helsinki, Finland | B107 (MKIII) | Double (no) | Laboratory measurements using 1000 Watt lamp, FMI facilities | $\sim$ 15 - 30 |
| Izaña, Tenerife, Spain | B185 (MKIII) | Double (no) | Measurements in climate chamber using 200 Watt lamp, PTB facilities | $\sim$ -5 - 50 |
| Lindenberg, Germany | B030 (MKIV) | Single (yes) | Measurements in climate chamber | $\sim$ 10 - 35 |
| | B078 (MKIV) | Single (yes) | using 200 Watt lamp, DWD facilities | $\sim$ 5 - 45 |
| Thessaloniki, Greece | B005 (MKII) | Single (yes) | Laboratory measurements | $\sim$ 15 - 50 |
| | B086 (MKIII) | Double (no) | using 1000 Watt lamp, LAP facilities | $\sim$ 15 - 45 |
| Sodankylä, Finland | B037 (MKII) | Single (yes) | Laboratory measurements | $\sim$ 10 - 45 |
| | B214 (MKIII) | Double (no) | using 1000 Watt lamp, FMI facilities | $\sim$ 10 -45 |