# Peer review of "Temperature dependence of the Brewer global UV measurements"

_Atmospheric Measurement Techniques, 2017_

## Referee Comment (RC1) · Anonymous Referee #1 · 12 Jul 2017

Brewer instruments were invented, designed and built to measure ozone column which is accomplished by means of ratio-metric measurement which to large extent is insensitive to instrument's optical throughput and PMT responsivity. The subsequent modification of the instrument to expand Brewer's application to measure irradiance seemed like a natural step, however was the stability of the instrument ever good enough to warrant it? This paper like several before it shows that w/o temperature corrections a long term monitoring of irradiance with Brewers will not produce data good enough to promote significant scientific discovery and precise RT model verifications. Perhaps it would be useful to provide a brief literature review highlighting the accomplishments (are there any?) of solar irradiance monitoring with Brewers.

To correct the temperature effect the instrument must be characterized. The instru-

ment can be treated like a black box w/o much understanding of how it operates and what is the mechanism of the temperature effect. One would expect that understanding the mechanism of the temperature effect would lead to a better methodology of temperature effect correction. Unfortunately, the characterization of Brewers fell on the shoulders of users who are forced to treat the instrument just as a black box because the manufacturers did not show much interest in solving the problem that in a better world would be their responsibility.

The thermal equilibrium of instrument with its environment does not imply that it is isothermal. Different parts of the instrument will have different temperatures. The question should be posed whether the parametrization of the black box with a single temperature is sufficient. Should it be the PMT's temperature or the ambient temperature? Obviously the best result will be obtained if both temperatures are used in the parametrization. If the two temperatures are correlated single temperature will suffice. However the perfect correlation is not the case because of the heater and heating from internal lamps, electronics and actuators. Plus there will be hysteresis.

In general case the temperature effect for each temperature range (TR1, TR2, TR3) should be modeled with the formula:

(1+A$\triangle$t_pmt)(1+B$\triangle$t_ambient)(1+a$\triangle$w+b($\triangle$w)ˆ2)

where $\triangle$w=w-w0 is wavelength increment form the reference wavelength w0.

You have decided to collapse the coefficients A and B into a single one and neglect the $\triangle$t_ambient. What is the cost of this approximation we will not know from your data. Authors should be commended for recognizing that the temperature effect is different in different temperature ranges (TR1, TR2 and TR3). In previous studies of Brewers by Cappellani and Kochler (1999), Weatherhead et al (2001), Garane et al. (2006) and Lakkala et al. (2008) this feature was not recognized. Probably it was because some of the studies used the 50 W lamps that heats up the diffuser which ends up masking the temperature effect of the diffuser. Still in Fig. 2 of Cappellani and Kochler (1999)

one can discern that the temperature coefficients are not constant through out the full range of temperatures. In Weatherhead et al (2001) data we see that for wavelengths greater than 325nm the temperature coefficients do not change with wavelengths and that all instruments have very similar (shapes) of temperature coefficients as function of wavelength differing only by wavelength independent offsets. This result may suggest that the wavelength dependence came from the nickel sulfate solar blind filter. But authors of the current study do not agree with it, right?

This confirms that the different patterns found between the three TRs are due to the change in the transmissivity of the Teflon diffuser.

This "this confirms" should be backed up with some illustration in this paper. Results of two tests: through diffuser and w/o diffuser.

For the measurements through the window it was found that the change in the response/°C is wavelength dependent for both the single and the double monochromator Brewers, indicating that the dependence of wavelength might not be introduced by the NiSO4 filter used only in the single monochromator Brewers as suggested by Garane et al. (2006).

Again this is speculative. Ylianttila and Schreder (2005) results suggest that Teflon introduces some wavelength dependence. The quantum efficiency of PMT's photocathode also has some temperature dependence that has a spectral component. Still nickel sulfite can't be acquitted from responsibility for wavelength dependence.

Anyway, Weatherhead et al (2001) results are not congruent with the present work.

I presume that each measurement was preceded with mercury scan to correct the wavelength shift due to temperature. It should be stated.

However correcting the wavelength shift at 297nm does not completely correct the wavelength shift at 325nm or 360nm. This wavelength shift is due to (1) translation of slits away from the optical axis, (2) diffraction grating grooves density change and (3)

micrometer screw expansion. The cumulative effect of wavelength change produces an apparent responsivity change, however this is not the true responsivity change as it depends on the spectral shape of the measured signal and it can't be applied to correct signal when measuring the solar spectral irradiance. It will be different for different lamps that have different spectral gradients dI(w)/dw, where I(w) is lamps irradiance. The authors should estimate this effect. BTW, I do not think anybody was concerned with this effect in previous works. This spurious effect due to wavelength shift may account for part of wavelength dependence in the measured effect. Keep in mind that manufacturer's claims on wavelength stability specs can't be trusted. What impact this study will have on Garane et al. (2006) and Lakkala et al. (2008)? The current results are not congruent with the previous results, right? The Fig 5 is offered as a degree of proof that the correction will improve data quality. I have several issues here: (a) Did you force the points for the red curves to be zero at t=25°C? It is too good to be just fortuitous.

(b) Why the "errors bars" in some case for blue (after correction) are wider than for red? This is counterintuitive. BTW, what do the "error bars" represent?

(c) The red curves should be closely approximated by the ratios of correction factors. I looked at the correction factors in the Supplement and looked at Fig. 3 and I do not get the same shapes as the red curves.

In Fig. 4 there is 315nm mentioned in the caption. It must be a mistake. The red colors for 005 and 037 are too similar. Make the vertical scales of panels a and be the same, i.e., say 0.7 units in each case. Frankly I do not understand panels c and d. 1-sigma of what? Do you have enough points to justify talking about statistics? This has to be explained and justified or dropped.

The paper is not easy to read. I had to go back and forth searching for info whether a given Brewer you were discussing is double or single and so on. I think a table with a list of Brewers, types, nickel sulfate yes or no, locations, temperature ranges, and

coefficients (from Supplement) would help. If you use the following formula

$(1+A\Delta t\_pmt)(1+a\Delta w+b(\Delta w)^2)$

the meaning and magnitude of coefficients A, a and b would be more easily readable. The coef A gives you general magnitude for w0 and a and b magnitude change with wavelength.

Also a method of measurements should be grouped in one place as apparently different Brewers were measured at different facilities with different equipment.

Fig. 3 shows that in some cases you did not have too many points. Also there are no data for TR1. Actually, what is the justification for having the same slopes for TR1 and TR3? You are paying here a price for ignoring the ambient temperature. I feel uneasy about TR2 width in some cases. If ranges are really due to Teflon they should be similar among instruments. The issue is that the big change of coefficients between TR2 and TR3 can't be explained by data from Ylianttila and Schreder (2005) and the change should occur in narrower range. Also that TR1 and TR3 are the same is not justified by Ylianttila and Schreder (2005).

I do understand that dealing with these instruments is a real pain. I understand that characterizing temperature effects is not an easy task particularly when you have no right equipment and facilities. So I am not surprised that the paper leaves many unanswered questions. Nevertheless I will recommend it being published providing that authors make some effort to fix and explain some issues that are within their reach. I feel sorry for the author they are forced to engage in such unsatisfactory endeavor.

---

## Referee Comment (RC2) · Anonymous Referee #2 · 14 Aug 2017

accept as is for the discussion phase

———————————————

---

## Referee Comment (RC3) · Anonymous Referee #2 · 14 Aug 2017

Brewer spectrophotometers have been used extensively in the past for measuring global UV irradiance. The high number of such instruments performing such measurements worldwide for very long periods, makes investigations like the one submitted valuable for the UV community.

Comments

Section 2

Since this paper is introducing a number of different instruments with different characteristics, including hardware, calibration facilities e.t.c. it would be essential for the authors to start this paragraph describing the main problem.

Temperature effect on Brewer measurements can be linked with PMT, diffuser differ-

ence in temperature compared with the ones during the standard calibration procedure, hysteresis effects, wavelength shift issues, other kind of stresses within the instrument.

The authors have to prioritize these effects and describe the methodology followed in order to eliminate or to investigate the effect of each of these factors.

A table with the instruments used together with some details on the method used for each one could be also useful for the reader.

Section 3

"carried inside" ? you mean moved from sun measurements to the calibration room ?

Figure 4 shows that changes are not wavelength independent, at least for some instruments

The standard deviations of what ? how many measurements have been performed for each temperature, wavelength ?

In addition it is worth noted that only one instrument has a positive change with temperature in TR3.

Are Ylliantilla results applicable to the Brewer using diffusers? Do some of the instruments use modified (that the Brewer initial) diffusers ?

Section 3.2

"This confirms that the different patterns found between the three TRs are due to the change in the transmissivity of the Teflon diffuser."

Based on this and figure 2: For measuring the instrument response the calibration is performed with the 1000W lamps in an environmental temperature of 25 degrees. Do this results imply that it is possible that, based on the above statement, the total duration of the normal calibration procedure may affect the calibration results ? (by having the diffuser heating up). Or most importantly that sun and 1000W-lamp measurements are partly incompatible due to the different diffuser temperature inside the calibration room and outside, even if the ambient and calibration room temperatures are 25 degrees ?

Figure 4. I wonder how reliable are the results for wavelengths below 300nm given the low counts that the instrument is measuring using the 1000W lamp.

Wavelength shifts for higher wavelengths is not out of the discussion even if a wavelength control/correction is performed at 297 (?) nm.

Conclusions

I understand that this is a mostly technical paper. But it would be interesting for the readers to demonstrate the brewer temperature effect a bit more clear and realistic. Reading this work I come to the conclusion that:

a. since there is a heater in the instrument (and not a cooling system), we are mainly interested by the temperature effects from 10 to 40+ degrees

b. Brewer temperature is different from ambient temperature (most probably the instrument (out on the sun) temperature is higher in the hot days than the temperature (measured in the shadow).

c. Temperature issues will affect mostly diurnal and seasonal global UV irradiance related investigations and not so much year to year trends.

So since the study is including a number of instruments that have different response in the temperature change and also perform measurements in very different environments concerning actual ambient (and Brewer) temperatures, it would be interesting to show the actual global UV % deviations for each site based on the actual brewer temperature. Either on a daily basis using the daily temperature change or during the year using the temperature at the measurement performed at the maximum -all year long existent- solar elevation angle.

---

## Referee Comment (RC4) · Anonymous Referee #3 · 22 Aug 2017

This paper describes work which is part of a larger collaboration to characterise and standardise operation, data processing and calibrations of, the Brewer Ozone Spectrophotometer, under the umbrella of COST Action ES1207. It is well written and illustrates clearly the problems with UV absolute calibration and makes a very good effort to provide a solution to significantly reduce the uncertainties which arise due to temperature dependence of the measurements.

It is also significant in that a Brewer network already exists which could provide a harmonised spectral UV network given sufficient study and rectification of the problems. In my opinion this paper provides a good starting point and should be published. However, I would hope that further studies will follow particularly with regard to long term stability in different climates and with different Brewer instruments.

---

## Author Comment (AC1) · 18 Oct 2017

**Response to Anonymous Referee #1**

After each comment by the referee (bold letters) there is the corresponding answer. The page and the line numbers referred correspond to the new version with tracked-changes.

**Brewer instruments were invented, designed and built to measure ozone column which is accomplished by means of ratio-metric measurement which to large extent is insensitive to instrument's optical throughput and PMT responsivity. The subsequent modification of the instrument to expand Brewer's application to measure irradiance seemed like a natural step, however was the stability of the instrument ever good enough to warrant it? This paper like several before it shows that w/o temperature corrections a long term monitoring of irradiance with Brewers will not produce data good enough to promote significant scientific discovery and precise RT model verifications. Perhaps it would be useful to provide a brief literature review highlighting the accomplishments (are there any?) of solar irradiance monitoring with Brewers.**

Reply

A literature review highlighting the usefulness of the spectral UV measurements from Brewers has been added in the introduction (P2,L15 – P3,L2).

**To correct the temperature effect the instrument must be characterized. The instrument can be treated like a black box w/o much understanding of how it operates and what is the mechanism of the temperature effect. One would expect that understanding the mechanism of the temperature effect would lead to a better methodology of temperature effect correction. Unfortunately, the characterization of Brewers fell on the shoulders of users who are forced to treat the instrument just as a black box because the manufacturers did not show much interest in solving the problem that in a better world would be their responsibility.**

Reply

The present study is focused on improving the quality of the spectral UV measurements by suggesting an improved characterization and correction methodology compared to those already existing. Obviously, no operator has the ability to characterize the instrument perfectly and fully explain the effect of temperature. Thus, several assumptions have been made in the study while uncertainties still exist. As shown by the study, all Brewers have an instrument specific response to the temperature dependence, but a common method applicable to the standard processing of the Brewer has to be derived to correct for the error introduced by the temperature dependence.

**The thermal equilibrium of instrument with its environment does not imply that it is isothermal. Different parts of the instrument will have different temperatures. The question should be posed whether the parametrization of the black box with a single temperature is sufficient. Should it be the PMT's temperature or the ambient temperature? Obviously the best result will be obtained if both temperatures are used in the parametrization. If the two temperatures are correlated single**

temperature will suffice. However the perfect correlation is not the case because of the heater and heating from internal lamps, electronics and actuators. Plus there will be hysteresis.

In general case the temperature effect for each temperature range (TR1, TR2, TR3) should be modeled with the formula:

$(1+A\Delta t\_pmt)(1+B\Delta t\_ambient)(1+a\Delta w+b(\Delta w)\text{^}2)$

where $\Delta w=w-w0$ is wavelength increment form the reference wavelength w0.

You have decided to collapse the coefficients A and B into a single one and neglect the $\Delta t\_ambient$. What is the cost of this approximation we will not know from your data.

Reply

It is sure that the parameterization of the black box with a single temperature is not sufficient. As the reviewer already mentioned, a parameterization including the external and the PMT temperature would lead to improved results, though it would not solve the problem since the temperature gradients inside the instrument do not depend solely on the difference between the external and the PMT temperatures. Nevertheless, the only available information for a standard user is the PMT temperature for each UV spectrum. A discussion regarding the used parameterization has been added in section 3.3 (P15, L6-12)

Authors should be commended for recognizing that the temperature effect is different in different temperature ranges (TR1, TR2 and TR3). In previous studies of Brewers by Cappellani and Kochler (1999), Weatherhead et al (2001), Garane et al. (2006) and Lakkala et al. (2008) this feature was not recognized. Probably it was because some of the studies used the 50 W lamps that heats up the diffuser which ends up masking the temperature effect of the diffuser. Still in Fig. 2 of Cappellani and Kochler (1999) one can discern that the temperature coefficients are not constant through out the full range of temperatures.

Reply

Even when a 1000 Watt lamp was used for the characterization (e.g. in Garane et al), there were only few data points below ~20°C which were possibly considered to be outliers.

In Weatherhead et al (2001) data we see that for wavelengths greater than 325nm the temperature coefficients do not change with wavelengths and that all instruments have very similar (shapes) of temperature coefficients as function of wavelength differing only by wavelength independent offsets. This result may suggest that the wavelength dependence came from the nickel sulfate solar blind filter. But authors of the current study do not agree with it, right?

Reply

We have not suggested that the nickel sulfate filter is not responsible for the dependence of wavelength. Indeed, we believe that the NiSO4 filter is responsible for part of the wavelength dependence (P12, L16-18). Changes have been made throughout the manuscript to make this clearer.

It is true that in Fig 1 of Weatherhead et al the results from most – not all - instruments are wavelength independent for wavelengths longer than 325 nm. However, there are instruments for which the dependence of wavelength is obvious (e.g. for Boulder, Everglades and Rocky Mountains).

Furthermore, analysis using a single trend for the entire range of temperatures may lead to misleading results, as, e.g., in Garane et al. (2006) where no wavelength dependence was found for B086. When the same results were re-analyzed by taking into account TR1, TR2 and TR3 the wavelength dependence was obvious.

Since part of the wavelength dependence is related to the transparency of the diffuser, the use of 50 Watt lamps by Weatherhead et al. may have also been responsible for not detecting any dependence of wavelength in their results. Additionally, the lamp tests for the study of Weatherhead et al were performed outside, leading to increased uncertainty in the results since the repeatability of measurements with 50W lamps is not very good due to the effect of changing environmental conditions (e.g., effect of the wind).

**This confirms that the different patterns found between the three TRs are due to the change in the transmissivity of the Teflon diffuser.**

**This "this confirms" should be backed up with some illustration in this paper. Results of two tests: through diffuser and w/o diffuser.**

Reply

Figure 5 and some relevant discussion were added in paragraph 3.2.

**For the measurements through the window it was found that the change in the response/∘C is wavelength dependent for both the single and the double monochromator Brewers, indicating that the dependence of wavelength might not be introduced by the NiSO4 filter used only in the single monochromator Brewers as suggested by Garane et al. (2006).**

**Again this is speculative. Ylianttila and Schreder (2005) results suggest that Teflon introduces some wavelength dependence. The quantum efficiency of PMT's photocathode also has some temperature dependence that has a spectral component. Still nickel sulfite can't be acquitted from responsibility for wavelength dependence.**

**Anyway, Weatherhead et al (2001) results are not congruent with the present work.**

Reply

What the reviewer says here is correct. The manuscript has been changed accordingly (P14, L5).

**I presume that each measurement was preceded with mercury scan to correct the wavelength shift due to temperature. It should be stated.**

Reply

The relevant information has been added in the manuscript (P10, L6-8).

**However correcting the wavelength shift at 297nm does not completely correct the wavelength shift at 325nm or 360nm. This wavelength shift is due to (1) translation of slits away from the optical axis, (2) diffraction grating grooves density change and (3) micrometer screw expansion. The cumulative effect of wavelength change produces an apparent responsivity change, however this is not the true responsivity change as it depends on the spectral shape of the measured signal and it can't be applied to correct signal when measuring the solar spectral irradiance. It will be different for different lamps that have different spectral gradients dI(w)/dw, where I(w) is lamps irradiance. The authors should estimate this effect. BTW, I do not think anybody was concerned with this effect in previous works. This spurious effect due to wavelength shift may account for part of wavelength dependence in the measured effect. Keep in mind that manufacturer's claims on wavelength stability specs can't be trusted. What impact this study will have on Garane et al. (2006) and Lakkala et al. (2008)? The current results are not congruent with the previous results, right?**

Reply

The effect of wavelength shift was studied for B005 and B086. The results are discussed in section 3.3.6.

**The Fig 5 is offered as a degree of proof that the correction will improve data quality. I have several issues here: (a) Did you force the points for the red curves to be zero at t=25◦C? It is too good to be just fortuitous.**

Reply

Yes, the points were forced to be zero at 25°C. Now it is also written in the legend of figure 6.

**(b) Why the "errors bars" in some cases for blue (after correction) are wider than for red? This is counterintuitive. BTW, what do the "error bars" represent?**

Reply

As it is now explained in the manuscript (P20, L10) the error bars represent the 1σ standard deviation of the mean for each 10°C bin. There was an error which caused the large differences in the original version of the manuscript (for the corrected data one more year of measurements was used). Small differences are

still evident in the revised version, since the correction is not the same for both instruments at each temperature.

**(c) The red curves should be closely approximated by the ratios of correction factors. I looked at the correction factors in the Supplement and looked at Fig. 3 and I do not get the same shapes as the red curves.**

Reply

This comment is again correct. The plotted ratio in figure 6a (5a in the previous version of the manuscript) was reversed. Now it is corrected and is very close to what would be expected based on the results presented in Fig 3.

**In Fig. 4 there is 315nm mentioned in the caption. It must be a mistake.**

Reply

It has been corrected.

**The red colors for 005 and 037 are too similar.**

Reply

The color of the line for B037 has been changed (Fig 3).

**Make the vertical scales of panels a and be the same, i.e., say 0.7 units in each case.**

Reply

Now they are both 0.75 units (Fig 3).

**Frankly I do not understand panels c and d. 1-sigma of what? Do you have enough points to justify talking about statistics? This has to be explained and justified or dropped.**

Reply

The comment was correct. Panels c and d have been removed.

**The paper is not easy to read. I had to go back and forth searching for info whether a given Brewer you were discussing is double or single and so on. I think a table with a list of Brewers, types, nickel sulfate yes or no, locations, temperature ranges, and coefficients (from Supplement) would help.**

Reply

Table 1 which contains all the suggested information has been added in Section 3.

**If you use the following formula**

$(1+A\Delta t\_pmt)( 1+a\Delta w+b(\Delta w)^2)$

**the meaning and magnitude of coefficients A, a and b would be more easily readable. The coef A gives you general magnitude for w0 and a and b magnitude change with wavelength.**

Reply

We think that the parameterization proposed here would be a bit more complicated to be reproduced by the Brewer operators after the instrument characterization. Thus we did not change the parameterization described in the supplement.

**Also a method of measurements should be grouped in one place as apparently different Brewers were measured at different facilities with different equipment.**

Reply

The relative information has been added in Table 1.

**Fig. 3 shows that in some cases you did not have too many points. Also there are no data for TR1. Actually, what is the justification for having the same slopes for TR1 and TR3?**

Reply

The (limited and uncertain) results that we have provide an indication that the slopes are not necessarily the same but similar (the differences are lower than the uncertainty of the derived slopes) in TR1 and TR3. Regarding the transmittance of the diffuser this assumption is verified in Fig 1 of Ylianttila and Schreder (2005). Measurements through the quartz window indicate that the different behavior in the three TRs is mainly due to the effect of temperature on the diffuser, thus we believe that this assumption is valid. Furthermore, the results presented in Figure 6 indicate that applying a correction based on this assumption improves the agreement between the instruments in TR1. Thus the correction is towards the right direction.

**You are paying here a price for ignoring the ambient temperature. I feel uneasy about TR2 width in some cases. If ranges are really due to Teflon they should be similar among instruments.**

Reply

The differences between the TR limits and ranges which were found for each case are estimated to be mostly related to the uncertainties in the characterization results and less to the characteristics of each instrument as it is now clearly stated in P8, L36. This issue is also discussed extensively in paragraph 3.3.

**The issue is that the big change of coefficients between TR2 and TR3 can't be explained by data from Ylianttila and Schreder (2005) and the change should occur in narrower range.**

Reply

In Figures 1-3 of Ylianttila and Schreder (2005) the change in TR2 ranges between 1 and 3% which is in agreement with what we found. Nevertheless in the same study it is stated that the change in transmissivity depends on the thickness of the used diffuser. Thus, if a particular Brewer is equipped with a thicker diffuser, compared to those used by Ylianttila and Schreder (2005), a larger change should be detected. The difference between the detected range of TR2 and the corresponding range detected by Ylianttila and Schreder (2005) is within the uncertainty limits described and justified in section 3.3.

**Also that TR1 and TR3 are the same is not justified by Ylianttila and Schreder (2005).**

Reply

This has been already answered above.

**I do understand that dealing with these instruments is a real pain. I understand that characterizing temperature effects is not an easy task particularly when you have no right equipment and facilities. So I am not surprised that the paper leaves many unanswered questions. Nevertheless I will recommend it being published providing that authors make some effort to fix and explain some issues that are within their reach. I feel sorry for the author they are forced to engage in such unsatisfactory endeavor.**

---

## Author Comment (AC2) · 18 Oct 2017

**Response to Anonymous Referee #2**

After each comment by the referee (bold letters) there is the corresponding answer. The page and the line numbers referred correspond to the new version with tracked-changes.

**Brewer spectrophotometers have been used extensively in the past for measuring global UV irradiance. The high number of such instruments performing such measurements worldwide for very long periods, makes investigations like the one submitted valuable for the UV community.**

 **Comments**

**Section 2**

**Since this paper is introducing a number of different instruments with different characteristics, including hardware, calibration facilities e.t.c. it would be essential for the authors to start this paragraph describing the main problem.**

Reply

The problem is now described clearly in the introduction (P3, L16-P4,L7).

**Temperature effect on Brewer measurements can be linked with PMT, diffuser difference in temperature compared with the ones during the standard calibration procedure, hysteresis effects, wavelength shift issues, other kind of stresses within the instrument.**

**The authors have to prioritize these effects and describe the methodology followed in order to eliminate or to investigate the effect of each of these factors.**

Reply

In the revised version of the manuscript, information regarding the complexity of this issue has been added in the introduction and the beginning of Sections 2 and 3. In the introduction of Section 3 there was already information regarding what has been done to avoid large temperature gradients inside the instruments and to ensure the good quality of the characterization procedure in each case. More information has been added in the new version regarding what has been done in order to eliminate the effect of wavelength shifts (P10, L6-8). Furthermore, a more detailed description of the remaining uncertainties is now provided in paragraph 3.3.

**A table with the instruments used together with some details on the method used for each one could be also useful for the reader.**

Reply

Table 1 has been added in Section 3, which contains all the suggested information.

**Section 3**

**"carried inside" ? you mean moved from sun measurements to the calibration room ?**

Reply

Yes, the manuscript has been changed so this is clear (P9, L11.

**Figure 4 shows that changes are not wavelength independent, at least for some instruments**

Reply

Yes, this is clearly stated in both, the old and the new version of the manuscript (P12, L14-18).

**The standard deviations of what ? how many measurements have been performed for each temperature, wavelength ?**

Reply

The reviewer has a point here. The discussion for the standard deviation might be misleading. Thus, the relative figures and discussion have been removed.

**In addition it is worth noted that only one instrument has a positive change with temperature in TR3.**

Reply

This information has been added in P12, L8.

**Are Ylliantilla results applicable to the Brewer using diffusers? Do some of the instruments use modified (that the Brewer initial) diffusers ?**

Reply

Discussion relative to this question has been added in section 3.2 (P14,L16 – 25)

**Section 3.2**

**"This confirms that the different patterns found between the three TRs are due to the change in the transmissivity of the Teflon diffuser."**

**Based on this and figure 2: For measuring the instrument response the calibration is performed with the 1000W lamps in an environmental temperature of 25 degrees. Do this results imply that it is possible that, based on the above statement, the total duration of the normal calibration procedure may affect the calibration results ? (by having the diffuser heating up). Or most**

**importantly that sun and 1000W-lamp measurements are partly incompatible due to the different diffuser temperature inside the calibration room and outside, even if the ambient and calibration room temperatures are 25 degrees ?**

Reply

Although the 1000 Watt lamps were not found to heat the diffuser (at least importantly), as was already discussed in section 2, the change of the room temperature during the characterization procedure may affect the results of the calibration. A more analytical discussion has been added in sub-section 3.3.3. As discussed in sub-section 3.3.1 the difference between the temperatures of the diffuser and the PMT is not important when the Brewer is illuminated by the sun. The relative uncertainties are already discussed in the same section.

**Figure 4. I wonder how reliable are the results for wavelengths below 300nm given the low counts that the instrument is measuring using the 1000W lamp.**

Reply

The level of the signal below 300 nm is of course lower than the signal at longer wavelengths, though for all the studied cases it is adequately high to give results of similar reliability to that of the results for longer wavelengths. For 1000 and Watt lamps the signal below 300 nm is much stronger than for the sun at the same wavelengths. We estimate that even the stray light effect on the measurements of single monochromator Brewers does not affect the results significantly.

**Wavelength shifts for higher wavelengths is not out of the discussion even if a wavelength control/correction is performed at 297 (?) nm.**

Reply

Relative discussion regarding the effects of wavelength shifts on the determination of the correction factors for the effect of temperature has been added (sub-section 3.3.6). Regarding the wavelength shift during field measurements, it may occur due to many reasons in addition to the effect of temperature. However, a wavelength correction is usually applied when the spectra are processed, which eliminates the greatest part of this later effect.

**Conclusions**

**I understand that this is a mostly technical paper. But it would be interesting for the readers to demonstrate the brewer temperature effect a bit more clear and realistic. Reading this work I come to the conclusion that:**

**a. since there is a heater in the instrument (and not a cooling system), we are mainly interested by the temperature effects from 10 to 40+ degrees**

**b. Brewer temperature is different from ambient temperature (most probably the instrument (out on the sun) temperature is higher in the hot days than the temperature (measured in the shadow).**

**c. Temperature issues will affect mostly diurnal and seasonal global UV irradiance related investigations and not so much year to year trends.**

Reply

All the above have been taken into account and relative discussion has been added (P21,L5 – P22,L4). However, (c) is partially true. Over high latitudes the mean temperature already changes, and is projected to change by more than 10 degrees until the end of the century. Such large changes might induce important biases in the study of the UV trends if the effect of temperature has not been removed from the spectra. Also there can be big differences (higher than 10 C degrees in mean temperature) between even subsequent years depending on the local weather conditions. The above are discussed in the paragraph of the Discussion and Conclusions section.

**So since the study is including a number of instruments that have different response in the temperature change and also perform measurements in very different environments concerning actual ambient (and Brewer) temperatures, it would be interesting to show the actual global UV % deviations for each site based on the actual brewer temperature. Either on a daily basis using the daily temperature change or during the year using the temperature at the measurement performed at the maximum -all year long existent- solar elevation angle.**

Reply

Figure 7 has been added in the manuscript.

---

## Author Comment (AC3) · 18 Oct 2017

We thank the referee #3 for his willingness to review this manuscript.

---

## Author Comment (AC4) · 18 Oct 2017

[revised manuscript text omitted]

---

## Author Comment (AC5) · 18 Oct 2017

[revised manuscript text omitted]